# An electrogenic redox loop in sulfate reduction reveals a likely widespread mechanism of energy conservation

Américo G. Duarte [1], Teresa Catarino[1,2], Gaye F. White[3], Diana Lousa[1], Sinje Neukirchen [4], Cláudio M. Soares[1], Filipa L. Sousa[4], Thomas A. Clarke[3] & Inês A.C. Pereira [1]

The bioenergetics of anaerobic metabolism frequently relies on redox loops performed by membrane complexes with substrate- and quinone-binding sites on opposite sides of the membrane. However, in sulfate respiration (a key process in the biogeochemical sulfur cycle), the substrate- and quinone-binding sites of the QrcABCD complex are periplasmic, and their role in energy conservation has not been elucidated. Here we show that the QrcABCD complex of *Desulfovibrio vulgaris* is electrogenic, as protons and electrons required for quinone reduction are extracted from opposite sides of the membrane, with a $H^+/e^-$ ratio of 1. Although the complex does not act as a $H^+$-pump, QrcD may include a conserved proton channel leading from the N-side to the P-side menaquinone pocket. Our work provides evidence of how energy is conserved during dissimilatory sulfate reduction, and suggests mechanisms behind the functions of related bacterial respiratory complexes in other bioenergetic contexts.

[1] Instituto de Tecnologia Química e Biológica António Xavier, Universidade Nova de Lisboa, Av. da República, 2780-157 Oeiras, Portugal. [2] Departamento de Química, Faculdade de Ciências e Tecnologia, Universidade Nova de Lisboa, 2829-516 Caparica, Portugal. [3] Centre for Molecular and Structural Biochemistry, School of Biological Sciences and School of Chemistry, University of East Anglia, Norwich NR4 7TJ, UK. [4] Division of Archaea Biology and Ecogenomics, Department of Ecogenomics and Systems Biology, University of Vienna, Althanstrasse 14 UZA I, 1090 Vienna, Austria. Correspondence and requests for materials should be addressed to I.A.C.P. (email: ipereira@itqb.unl.pt)

The dissimilatory reduction of sulfate is one of the most important processes of microbial respiration in anoxic environments, where it plays a major role in both the sulfur and carbon biogeochemical cycles[1,2]. This process is particularly relevant in marine sediments where it is estimated that 11.3 Tmol of sulfate is reduced per year, accounting for up to 50% of the total organic carbon remineralization in the sea floor. It has become increasingly recognized that dissimilatory sulfate reduction is also important in low-sulfate environments fueled by a cryptic sulfur cycle[3,4], where it plays an important role in mitigating methane emissions through anaerobic methane oxidation[5]. Despite its global importance, it is still not clear how microorganisms can conserve energy from the reduction of sulfate, which requires an ATP investment for its activation[6]. Three respiratory membrane complexes are widely conserved in sulfate reducers and seem to be involved in energy conservation: QrcABCD[7,8], QmoABC[9,10], and DsrMKJOP[11,12]. The Qrc complex has been shown to oxidize its physiological electron donor, the soluble tetraheme cytochrome $c_3$ (TpI$c_3$), and reduce the menaquinone (MK) pool[7] (Fig. 1), whereas both the Qmo and Dsr complexes are proposed to be involved in electron transfer to enzymes of the soluble sulfate reduction pathway[1,13]. Nevertheless, it is not known how electron transfer through these membrane complexes contributes to the generation of a chemiosmotic gradient during sulfate reduction.

The architecture of the Qrc, Qmo, and Dsr complexes follows the typical modular character of prokaryotic multi-subunit redox proteins, where different arrangements of redox modules are associated with different functionalities[14,15]. The QrcABCD complex is a striking example, being formed by the association of a membrane-anchored multiheme cytochrome $c$ (QrcA) with three subunits (QrcBCD) related with subunits of molybdo/tungsten oxidoreductases of the Mo/W-bis-pterin guanosine

dinucleotide (Mo/W-bisPGD) family[16,17]. QrcB is homologous to the catalytic subunits of this family, but does not contain a Mo/W-pterin cofactor, while QrcC is homologous to the electron transfer subunits containing four Fe–S clusters, and QrcD to the integral membrane subunits that interact with quinones and have no cofactors[7]. The QrcABC proteins are on the periplasmic side of the cytoplasmic membrane and this complex links periplasmic hydrogen and formate oxidation to the MK pool[7,8]. It is present in the large family of Deltaproteobacteria sulfate reducers, which are characterized by an abundance of multiheme cytochromes $c$ (namely TpI$c_3$), and hydrogenases or formate dehydrogenases that lack a membrane subunit for direct quinone reduction[6]. In *Desulfovibrio alaskensis* G20, the Qrc complex was shown to be essential for growth on $H_2$ or formate and sulfate[8] and it has also been implicated in syntrophic growth[18,19].

The key protein in the possible electrogenic role of the Qrc complex is the QrcD subunit. This is an integral membrane protein that belongs to the so-called NrfD/PsrC family, which comprises bacterial proteins with 8–10 TMH that interact with quinones but have no redox cofactors[20]. Proteins in this family always associate with an electron transfer subunit containing up to four iron–sulfur clusters, forming a quinol dehydrogenase/quinone reductase dimeric redox module that may be part of a complex with one catalytic or further subunits (as in the Qrc complex) or operate alone transferring electrons to soluble periplasmic proteins. Examples of this dimeric module are widespread in bacterial respiratory chains, including in nitrite reduction (NrfCD)[21], polysulfide reduction (PsrBC)[22,23], tetrathionate reduction (TtrBC)[24], DMSO reduction (DmsBC)[25], sulfur reduction (SreBC)[26], sulfite reduction (MccCD)[27], arsenate reduction (ArrBC)[28], oxygen reduction (ActBC)[29], sulfate reduction (DsrOP and HmcDC)[9,30], hydrogen oxidation/production (HybAB)[31], sulfite oxidation (SoeBC)[32], and hydrogen/formate oxidation/production in the case of QrcCD (Supplementary Table 1 and Supplementary Fig. 1). This widespread distribution among bacterial anaerobic respiratory chains strongly suggests an ancient origin for this redox module. Structural information on this module is available through the structures of the *Thermus thermophilus* PsrABC polysulfide reductase-like enzyme[23] (here named as PsrABC2, since it may actually not be a polysulfide reductase—see below), and recently of the alternative complex III (ActABCDEF)[29,33]. Both structures confirm the absence of cofactors and the presence of a quinone binding site in the integral membrane subunit, which is located close to the periplasmic side of the membrane and in short distance to the proximal iron–sulfur cluster of the electron transfer subunit. A similar location for the quinone binding site was observed in QrcD, through the effect of a quinol analogue on the EPR spectrum of the proximal [3Fe–4S] cluster[7,34]. Despite the widespread occurrence of this dimeric redox module in bacterial respiratory chains it has still not been elucidated whether it is associated with energy conservation. These proteins have been assumed to be electroneutral[35] due to the location of the quinone-binding site on the same side of the membrane as the electron input/output, and because in some cases thermodynamics precludes energy conservation. However, PsrC2[23], HybB[31], and ActC[29,33] have been proposed to act as conformational proton pumps, but still without experimental validation.

Here, we report the reconstitution of the *Desulfovibrio vulgaris* Hildenborough QrcABCD membrane complex in liposomes and show that the complex is electrogenic, as protons and electrons required for MK reduction are extracted from opposite sides of the membrane (Fig. 1). However, it does not act as a $H^+$-pump. A model for the QrcD subunit allowed the identification of a conserved proton channel leading from the N-side up to the MK pocket. Sequence and phylogenetic analyses indicate that QrcD is

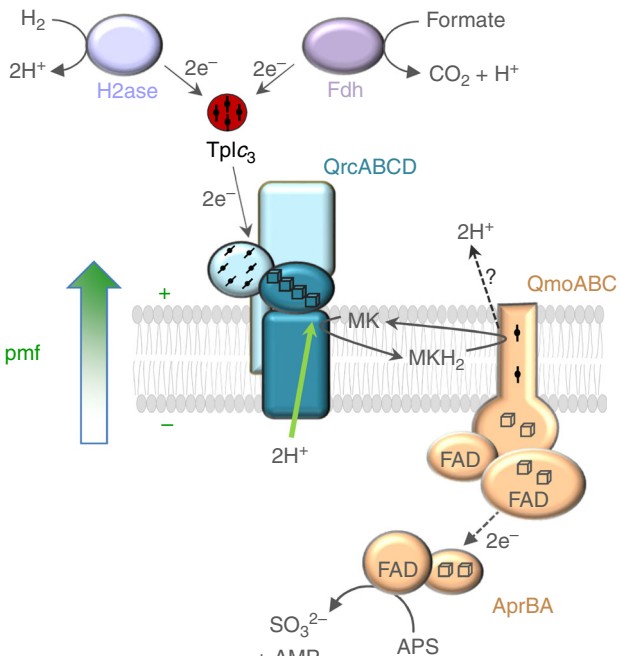

**Fig. 1** Physiological role of the QrcABCD complex. The QrcABCD complex accepts two electrons from the TpI$c_3$ and reduces MK. We show here that uptake of $2H^+$ for MK reduction occurs from the cytoplasm, thereby contributing to the *pmf*. MKH$_2$ may be oxidized by the QmoABC complex. The QrcCD dimeric redox module is highlighted by a darker shade of blue. H2ase hydrogenase, Fdh formate dehydrogenase, APS adenosine 5′-phosphosulfate, AprAB adenosine 5′-phosphosulfate reductase

closely related to HybB and HmcC, which are likely also electrogenic. This work provides the first evidence of how energy is conserved during the dissimilatory reduction of sulfate, and how this ancient and widespread dimeric redox module may operate in other bacterial respiratory chains.

## Results

**Characterization of QrcABCD liposomes**. The QrcABCD membrane complex was reconstituted in liposomes (Supplementary Fig. 2), in the presence or absence of menaquinone-4 (MK4), according to the onset method[36]. Four types of liposomes were used in each experiment: proteoliposomes containing Qrc and MK4, proteoliposomes containing only Qrc, liposomes containing only MK4 and empty liposomes. The vesicle integrity was tested through the generation of a membrane potential in the presence of valinomycin (a $K^+$-specific ionophore), $K^+$ ions, and oxonol VI as membrane potential probe. The absorbance of this anionic dye increases when an inside-positive potential is developed[37,38]. A positive shift in the absorption was observed for all liposome preparations when KCl was added, due to the membrane potential created from the $K^+$ influx facilitated by valinomycin[38], confirming that the liposomes are closed (Supplementary Fig. 3).

Upon reconstitution, the Qrc complex is expected to insert in the liposomes with its large hydrophilic head, formed by the QrcABC subunits, oriented mainly to the outside, as the opposite orientation would require overcoming the energetic barrier of translocating these subunits across the partially-open bilayer. To evaluate this orientation, the Qrc liposomes were treated with proteinase K, followed by SDS-PAGE analysis coupled with heme-staining to detect the heme-containing QrcA subunit, as described[39]. After incubation with proteinase K, the QrcA subunit is hydrolyzed and can no longer be detected (Supplementary Fig. 4). In a control experiment with a soluble cytochrome $c$ present inside the liposomes no degradation was observed. This confirms that the QrcABCD complex inserts into

the vesicles with its soluble subunits mainly facing the outside solution. Thus, the liposome outside solution mimics the bacterial periplasm and the inner compartment the cytoplasm.

**Electron transfer and generation of a membrane potential**. The physiological electron donor of the Qrc complex is the $TpIc_3$ cytochrome. For all electron transfer experiments, this cytochrome was reduced only to about 90% to make sure there was no excess of reductant present. Reduced $TpIc_3$ was mixed with each of the four different types of liposomes, in a stopped-flow apparatus, and electron transfer (ET) from the cytochrome was measured. A high re-oxidation rate of $TpIc_3$ is observed with the Qrc/MK4 proteoliposomes (Fig. 2a), whereas it is negligible in the absence of MK4 (confirming there are no traces of $O_2$). Some re-oxidation is also observed with MK4-only liposomes, due to direct ET between reduced $TpIc_3$ and MK4. This unspecific process is facilitated by the interaction of the positively-charged $TpIc_3$ with the negative lipid groups, but occurs with a much slower rate and to a less extent than for Qrc/MK4 liposomes. Since the experiments were performed under steady-state conditions with identical concentrations of liposomes/MK4, it is possible to subtract the unspecific re-oxidation rate measured with MK4-liposomes to determine the specific $TpIc_3$ re-oxidation rate catalyzed by Qrc (Fig. 2b). The difference trace is well fitted with a single exponential decay, with a rate constant ($k$) of $0.65 \pm 0.05\ s^{-1}$, where $\pm$ gives the standard deviation error obtained from the fit.

To determine if a membrane potential is generated during ET, the experiments were repeated in the presence of oxonol V. Liposomes from the same preparations were used, both in the absence and in the presence of valinomycin, which permeabilises the liposomes and dissipates the membrane potential. All traces are superimposable in the presence of valinomycin (Supplementary Fig. 5) and to observe the actual membrane potential detected by oxonol V we subtracted the traces with valinomycin from those without valinomycin (Fig. 2c, see also Supplementary

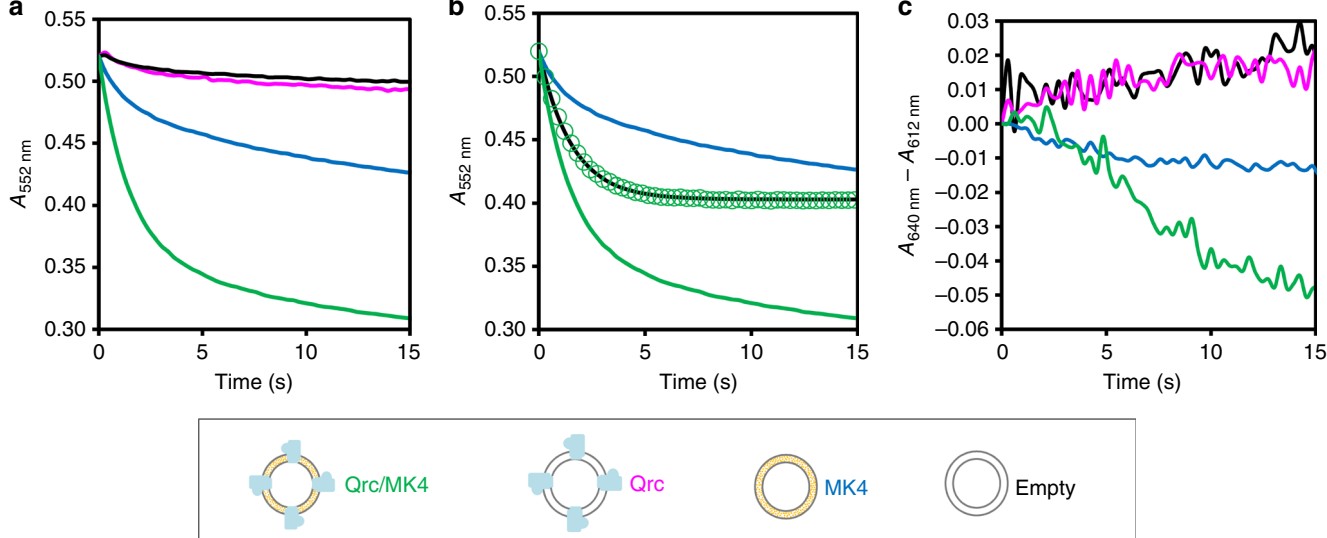

**Fig. 2** Electron transfer and membrane potential generation. **a** Absorbance changes at 552 nm after mixing reduced $TpIc_3$ with empty liposomes (black), Qrc proteoliposomes (without MK4) (magenta), MK4-liposomes (blue), and Qrc/MK4 proteoliposomes (green). **b** Kinetic trace subtraction (green circles) of MK4-liposomes data (blue) from that of Qrc/MK4 proteoliposomes (green), fitted with a single exponential (black line) with equation: $(-0.12 \pm 0.01) \times \exp^{(0.65\pm0.05)\times t} + (0.40 \pm 0.03)$, where $\pm$ gives the standard deviation error obtained from the fit for each value. **c** Changes in $A_{640\,nm} - A_{612\,nm}$ after mixing reduced $TpIc_3$ with liposomes in the presence of oxonol V. The absorbance presented is the subtraction between liposomes without and with valinomycin (traces used for the subtractions are presented in Supplementary Fig. 6): empty liposomes (black), Qrc proteoliposomes (magenta), MK4-liposomes (blue), and Qrc/MK4 proteoliposomes (green). All traces correspond to the average of at least seven experiments performed with at least two distinct liposome preparations

Fig. 6 for the raw data). The Qrc/MK4 proteoliposomes present a clear negative shift of the measured absorbance, corresponding to the development of an inside negative potential. This potential develops in the same time frame as ET (but with a slower rate because oxonol V is a slow-responding dye[37]), indicating that this process creates a charge separation across the membrane, with the inner solution having an excess of negative charges. Empty liposomes and Qrc proteoliposomes present a residual increase in absorbance possibly reflecting a small influx of $K^+$ ions to compensate the positively charged $TpIc_3$, whereas with MK4 liposomes the residual decrease in absorbance might result from positive charge leakage due to the external consumption of protons in the non-specific reduction of MK4.

**Direction of proton uptake**. To investigate the directionality of proton uptake by the Qrc complex, we prepared liposomes with phenol red trapped inside the vesicles. Phenol red is a pH-sensitive dye whose absorbance at 555 nm increases when deprotonated, and which is frequently used to monitor proton uptake in proteoliposomes[40–42]. The vesicle integrity was evaluated as described above (Supplementary Fig. 3). To avoid inhibition of proton uptake, valinomycin was also incorporated in the liposomes to ensure dissipation of the membrane potential created during MK4 reduction. The ET observed (Fig. 3a) was

similar to that previously observed in the absence of valinomycin (Fig. 2a). The protonation state of phenol red inside the vesicles was monitored at 560 nm, which is an isosbestic point in the $TpIc_3$ absorbance spectrum where redox changes have a minimal contribution. For the Qrc/MK4 liposomes, an exponential increase in phenol red absorbance is detected upon ET (Fig. 3b), consistent with proton uptake from the inner solution. This absorbance increase occurs in the same time frame as the $TpIc_3$ re-oxidation (Fig. 3a), indicating simultaneous ET and proton consumption. Experiments with MK4 liposomes lacking the Qrc complex show a small linear increase in absorbance, attributed to some proton leakage from the inner solution upon unspecific reduction of MK4 and corresponding $H^+$ uptake from the outer solution. The liposomes without MK4 present negligible variations in phenol red absorbance. These results reveal that the protons required for MK4 reduction by QrcD are taken up from the liposomes inner solution.

Next we performed similar experiments but with phenol red in the outer solution. The ET traces (Fig. 3c) are again similar, whereas for phenol red absorption the Qrc/MK4 proteoliposomes now show a similar behavior to the MK4 liposomes up to 5 s, remaining stable after that (Fig. 3d). The increase in absorbance is due to the uptake of protons from the outer solution for the unspecific reduction of MK4 by $TpIc_3$, and the results show that

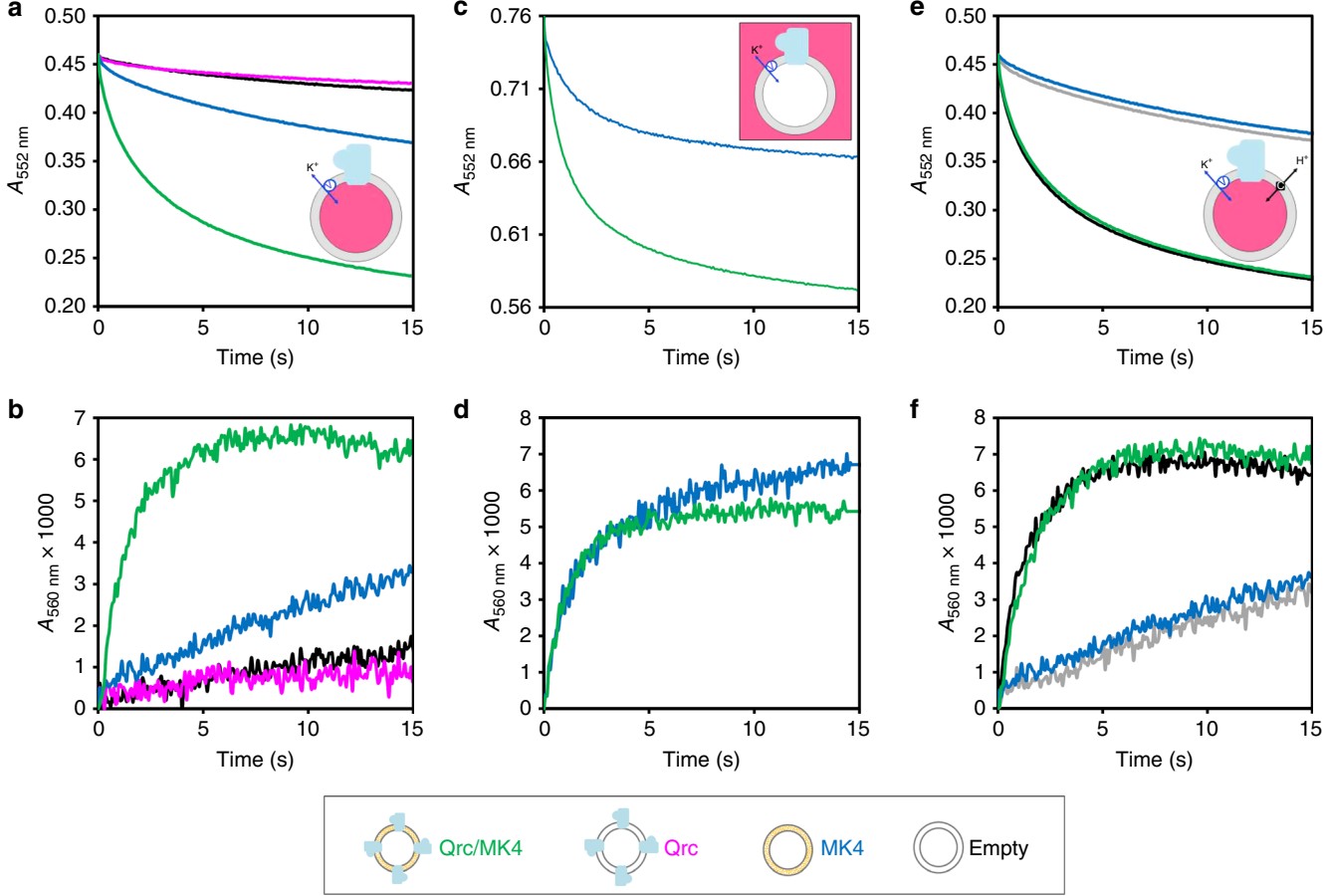

**Fig. 3** Electron and proton transfer. **a** Absorbance changes at 552 nm and **b** at 560 nm, after mixing reduced TpIc₃ with valinomycin permeabilised liposomes prepared with phenol red in the inner solution. Empty liposomes (black), Qrc proteoliposomes (magenta), MK4-liposomes (blue), and Qrc/MK4 proteoliposomes (green). **c** Absorbance changes at 552 nm and **d** at 560 nm, after mixing reduced Tplc₃ with valinomycin permeabilised liposomes and phenol red in the outer solution, MK4-liposomes (blue) and Qrc/MK4 proteoliposomes (green). **e** Absorbance changes at 552 nm and **f** at 560 nm, after mixing reduced Tplc₃ with valinomycin permeabilised liposomes prepared with phenol red in the inner solution in the presence of CCCP, MK4-liposomes (blue), Qrc/MK4 proteoliposomes (green), CCCP permeabilised MK4-liposomes (grey), and CCCP permeabilised Qrc/MK4 proteoliposomes (black). All traces correspond to the average of at least seven experiments performed with at least two distinct liposome preparations

the Qrc complex has no effect on this proton uptake, ruling out proton pumping activity. The trace for the Qrc/MK4 proteoliposomes stabilizes after about 5 s, whereas for the MK4-only liposomes a small linear increase is still observed after that. This is because at 5 s the concentration of oxidized quinones is still significant in the MK4-only experiment but is already negligible in the experiment with Qrc/MK4 proteoliposomes, due to their fast reduction by the complex. Importantly, this experiment reveals that no protons are pumped outside the proteoliposomes during ET from Qrc to MK4.

To further confirm this, we performed experiments again with phenol red in the inner compartment in the presence of valinomycin, with or without the protonophore carbonyl cyanide m-chlorophenyl hydrazine (CCCP). The ET is not affected by the presence or absence of CCCP (Fig. 3e), and the same is observed for the phenol red absorbance (Fig. 3f), indicating similar proton consumption from the inner solution with or without CCCP. These experiments reveal that the protons taken up from inside the liposomes are scalar protons consumed in the reduction of MK4 by Qrc, and confirm that there is no proton pumping associated with this reaction. A similar effect was observed for the scalar protons consumed by cytochrome $c$ oxidase[41] or by Complex I[43].

**Proton–electron ratio**. To determine the proton/electron ratio of the Qrc complex, we converted the absorbance changes of TpI$c_3$ to reduced heme concentration, while the absorbance of phenol red, was calibrated with the irreversible tryptic hydrolysis of N-α-tosyl-L-arginyl-O-methylester (TAME) (Supplementary Fig. 7). We converted the subtracted data from Fig. 3b to concentration of reduced heme, which is shown alongside with concentration of protons consumed (Fig. 4). The heme oxidation and proton consumption follow inverse single exponentials, which are fitted with similar kinetic parameters: $k^{Fe} = 0.60 \pm 0.05\ s^{-1}$ and $A^{Fe} = 5.03 \pm 0.30\ \mu M$ for heme $Fe^{2+}$ oxidation and $k^{H} = 0.68 \pm 0.06\ s^{-1}$ and $A^{H} = -5.22 \pm 0.26\ \mu M$ for $H^+$ consumption. This indicates a $H^+/e^-$ ratio of 1 for the Qrc complex, determined either by the ratio of rate constants or exponential amplitudes. The observed

rate constant ($k^{Fe} = 0.60 \pm 0.05\ s^{-1}$) can be used to calculate the Qrc turnover number (see Methods and Supplementary Fig. 8). The value obtained, $147\ s^{-1}$, is much higher than the value previously observed for the Qrc complex solubilized in detergent ($0.28\ s^{-1}$)[7], most likely because the liposome experiments more closely resemble the physiological environment.

The concentration of protons taken up from the outside solution for the unspecific reduction of MK4 by the cytochrome, as well as the concentration of heme oxidized in the process was also calculated from Fig. 3c, d, in a similar manner. The results (Supplementary Fig. 9), reveal that about 4 μM of protons are consumed outside, which is within the range that can be detected by phenol red (Supplementary Fig. 7). As expected, the proton/electron ratio of the unspecific process, as determined from the fit of the experimental data, is also 1.

**QrcCD structural model**. The quinone binding site of Qrc was previously shown to be located on the periplasmic side of QrcD, and close to the [3Fe–4S] cluster of QrcC[7]. A similar periplasmic orientation of the quinone binding site was observed in the structure of PsrC[23], and predicted in the structures of ActC[29,33]. Since proton uptake is observed from the opposite side of the membrane in the Qrc complex, this suggests the presence of a proton channel in QrcD. To get further insight into this we generated a homology model for the QrcCD module, based on the structure of *Rhodothermus marinus* ActC (homologous with QrcD) and C-terminus of ActB[29] (ActB$_C$, homologous with QrcC). ActC is the closest relative to QrcD that has been structurally characterized and is composed of ten transmembrane helices (TMH), organized in two four-helix bundles and one helix dimer, as also predicted for QrcD. The only other member of this protein family that has been structurally characterized is PsrC2[23], but this has only eight TMH. The generated QrcD model contains ten TMH that superimpose very well with the corresponding ActC structure (Fig. 5a). We identified a cavity in the QrcD model that is the likely MK-binding site, containing several residues strictly conserved in the QrcD family, namely F63, I67, D120, M139, L140, and E142 (Fig. 5b and Supplementary Fig. 10). Importantly, this MK-binding site is shielded from direct contact with the periplasm by the QrcC subunit, likely preventing $H^+$ uptake from the periplasmic side of the membrane. In agreement with the previous EPR studies[7], the MK-binding site is very close to the predicted [3Fe–4S] cluster of the QrcC subunit. Interestingly, we identify residue Y113 from QrcC as likely participating also in the MK-binding site, and this residue is conserved both in QrcC and ActB proteins. The proposed quinol binding pocket in the ActC structure[29] is located in the same region and several of the residues are also conserved between the QrcD and ActC.

In addition, we identified a chain of polar residues in the QrcD model, forming a pathway that spans from the cytoplasmic side of QrcD towards the predicted MK-binding site (Fig. 5b). These residues include D70, Y88, N100, Y110, Y150, E157, E164, H199, and E413 (*D. vulgaris* QrcD numbering), which are all strictly conserved in the QrcD family (Supplementary Fig. 10), and form the possible $H^+$ transfer pathway by which $H^+$ from the cytoplasm reach the MK-binding site. The first residue in this pathway close to the MK site, D70, corresponds to Y23 of *Wolinella succinogenes* PsrC, which was shown to be essential for activity[22]. The distance between Y110 and D70 (8.8 Å), and Y150 and E157 (9.9 Å) is likely bridged by water molecules present inside the channel.

The proposed QrcD channel is in the same region as the cytoplasmic half channel proposed for ActC (Supplementary Fig. 11), but most of the protonable residues are not conserved nor in the same positions, with the exception of Y150 (QrcD

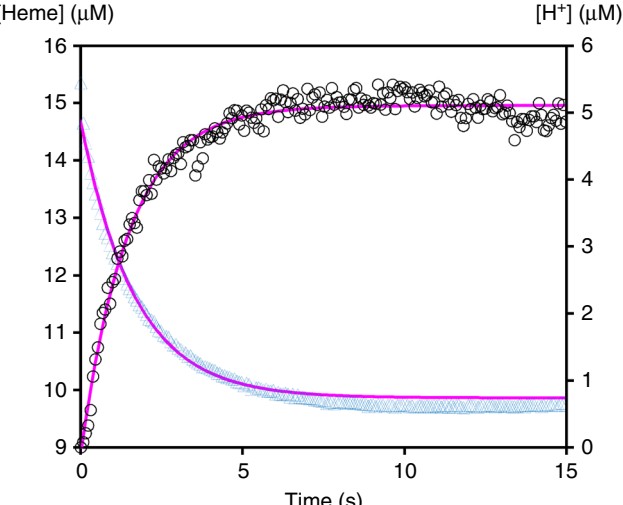

**Fig. 4** Electron transfer and proton consumption in Qrc/MK4 proteoliposomes. Ferrous heme concentration (blue triangles) and proton concentration (black circles) after mixture of reduced TpI$c_3$ and Qrc/MK4 liposomes (with phenol red inside). The magenta lines correspond to single exponential fits with the following equations: heme oxidation: $(5.03 \pm 0.30) \times exp^{(-0.60 \pm 0.05) \times t} + (9.78 \pm 0.40)$; proton consumption: $-(5.22 \pm 0.26) \times exp^{(-0.68 \pm 0.06) \times t} + (5.11 \pm 0.15)$

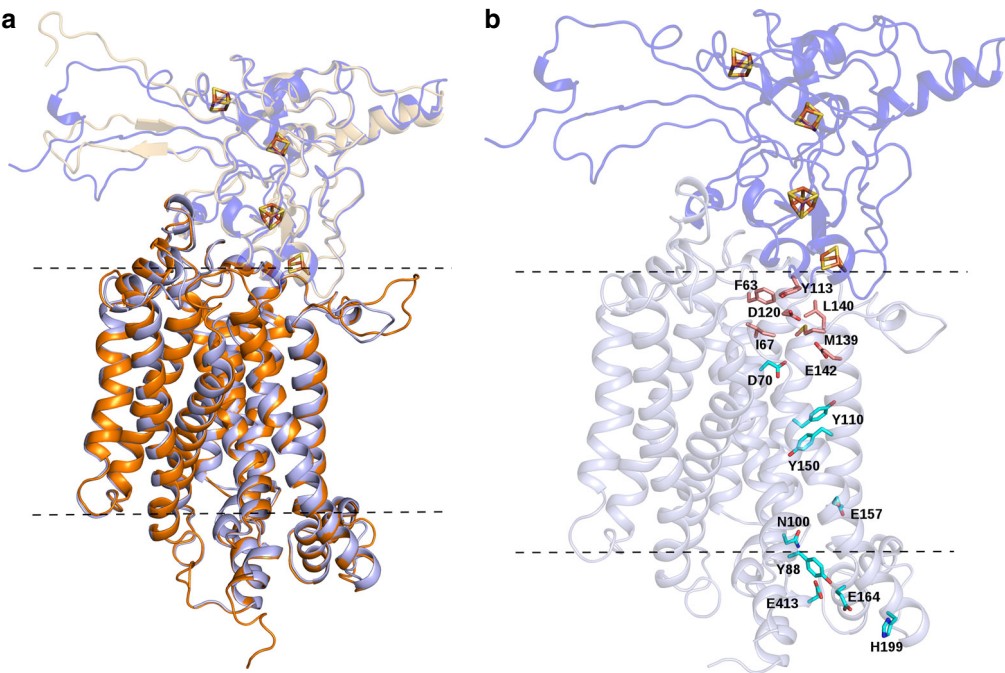

**Fig. 5** QrcCD homology-based model. **a** The structural model of Qrc subunits C (QrcC, purple with transparency) and D (QrcD, light purple) is superimposed on the structure of subunits B C-terminus (ActB, beige with transparency) and C (ActC, orange) of the alternative complex III (PDB code: 6FOK[29]). A cartoon representation is used to depict the backbone structure of the two proteins and the iron–sulfur clusters are represented by sticks, with iron and sulfur atoms colored in orange and yellow, respectively. **b** The Qrc residues that are putatively involved in quinone binding and proton translocation are highlighted using sticks with carbons colored in pink and cyan, respectively. The approximate membrane boundaries are depicted in dashed lines

numbering), which is conserved between the two proteins. In QrcD the proton pathway is connected to the quinone-binding site by D70, whereas in the case of the ActC cytoplasmic half-channel there is apparently no connection to the quinone-binding site. In contrast, ActC is proposed to have another half-channel connecting to the periplasmic side[29], which we could not observe in QrcD. These findings agree with the suggestion that the two proteins evolved separately, from a common ancestor, to have different mechanisms of energy conservation (see below).

**Phylogenetic analysis of the QrcD/NrfD/PsrC family.** In order to investigate the evolutionary history and relationship between members of the bacterial QrcD/PsrC/NrfD family, we retrieved sequences for homologous genes to each of the families from available genomes, taking into account only cases found next to the cognate iron–sulfur protein and presenting the typical gene cluster arrangement of each family. Initially a pairwise sequence identity analysis between all sequences was performed, using a threshold of 25% local identity ($E$-value $< 10^{-10}$) (Supplementary Fig. 12). Curiously, we found two groups of PsrC proteins, one represented by the *W. succinogenes* PsrC[22] that we called PsrC1, and the other represented by the PsrC of *T. thermophilus*, for which the crystal structure has been determined[23], that we called PsrC2. The PsrC2 family does not show identities above 25% with any other protein family and form a separate cluster, and the same is also observed for SreC, DmsC, and SoeC. Thus, the SoeC, DmsC, PsrC2, and SreC sequences were excluded from the phylogenetic analysis to avoid potential unreliable results and/or long-branch attraction artifacts resulting from the low level of sequence identity. The exclusion of these families does not per se exclude their homologous relationships. In our view, not only do these proteins share structural homology, but most likely, also evolutionary ancestry. However, due to the lack of cofactors, through time the evolutionary pressure acted upon structural

conservation instead of specific amino acid conservation, which renders their inclusion in any phylogenetic analysis misleading regarding interfamily relationships.

We reconstructed a maximum-likelihood phylogeny for the other families (Fig. 6), based upon an alignment calculated in Praline, which combines an iterative PSI-BLAST approach with secondary TMH predictions. In this phylogeny, there is a separation of the 10 TMH-containing protein families from the ones with 8 or 9 TMH. The tetrathionate reductase TtrC clade (nine TMH) is the most basal member of this second clade, which may agree with the suggestion that this enzyme originated prior to the Bacteria/Archaea divide[28], and that the dimeric redox module is of ancient origin. As might be expected, the proteins containing ten TMH (QrcD, HybB, HmcC, DsrP, and ActC) are more closely related and seem to have diverged from a protein related to the ancestor of the PsrC1, NrfD, and MccD group, which cluster together with ArrC and TtrC. This arrangement suggests an evolutionary event that was associated with the addition of two extra TMH to an ancestral module containing 8 TMH. Curiously, the group of proteins with ten TMH is associated with "catalytic" subunits that are not of the standard Mo/W-*bis*PGD enzyme family, and which include the cofactor-less QrcB protein (albeit homologous to the Mo/W-*bis*PGD catalytic subunits), the 16-heme cytochrome HmcA, the [NiFe] hydrogenase subunit HybA, the triheme cytochrome DsrJ and finally ActB, which is a fusion of a cofactor-less Mo/W-*bis*PGD-like protein with the iron–sulfur subunit. The Qrc complex has in common with the Act complex not only the presence of a cofactor-less Mo/W-*bis*PGD-like protein, but also the presence of an additional penta/hexaheme cytochrome subunit (QrcA/ActA), which could suggest a close relationship. However, the QrcD and ActC proteins are found in separate clades, although they probably share a common precursor. This difference may be related with the fact that they catalyze opposite reactions, as Qrc

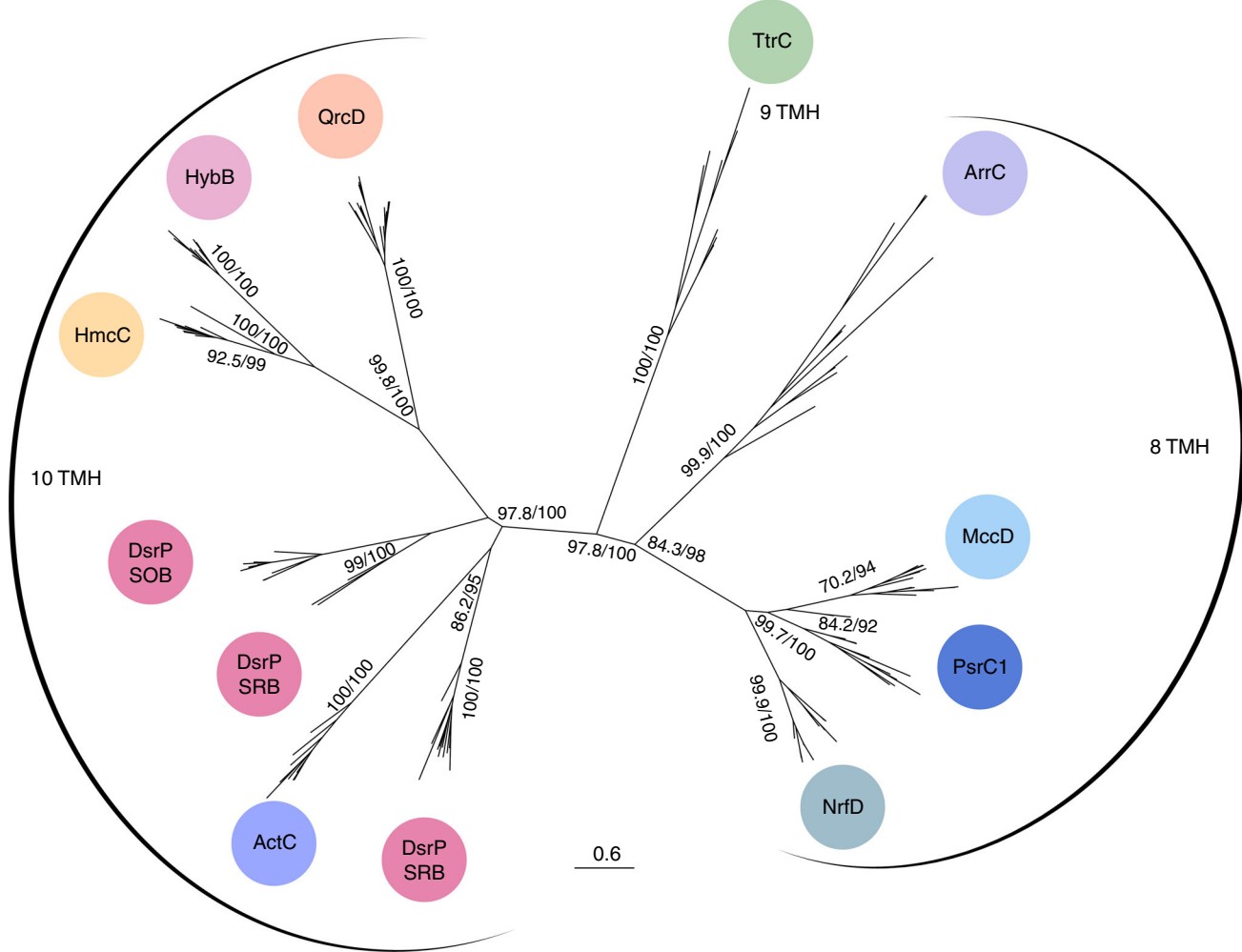

**Fig. 6** Maximum-likelihood phylogeny of the QrcC/PsrC/NrfD family. Node numbers represent bootstrap support values evaluated with -alrt and with 1000 bootstrap replicates, respectively. Only high support bootstraps values (>80) of the major splits are shown. Abbreviations according to the text. SRB sulfate reducing bacteria, SOB sulfur oxidizing bacteria, SRA sulfate reducing archaea (*Archaeoglobus* spp.). Of note *A. sulfaticallidus* DsrP clusters with the bacterial DsrP proteins

oxidizes a cytochrome *c* and reduces menaquinone, whereas Act oxidizes menaquinol and reduces a cytochrome *c*, and agrees with their apparently different mode of energy conservation.

So far, QrcD was found only in the deltaproteobacteria, indicating the appearance of the Qrc complex in this class, probably evolving from a Mo/W-*bis*PGD enzyme precursor to the Psr1, Nrf, and Mcc systems. Interestingly, the QrcD clade is found close to another quinone reductase, HybB, and to HmcC, suggesting that the Hmc complex may also act preferably in the direction of quinone reduction. In HybB and HmcC, most of the residues proposed to form the quinone binding site and the H$^+$ transfer pathway in QrcD are conserved (Supplementary Figs. 13 and 14), suggesting they may also be electrogenic.

## Discussion

In cellular respiration, energy is transduced through conversion of the free energy of a redox reaction into an electrochemical ion gradient across a membrane (the proton or sodium motive force), which is used to drive the synthesis of ATP through ATP synthases. There are two fundamental mechanisms by which ion translocation is achieved, namely proton pumping or redox loops[35,44]. In anaerobic respiration, where the redox conversions involve lower free energy changes than in aerobic respiration, the

redox loop mechanism is most commonly observed. In this mechanism, ion translocation is established through charge separation performed by quinone-reacting membrane protein complexes that usually have substrate and quinone binding sites on opposite sides of the membrane. The charge separation is achieved as the result of electron transfer from the positive (P) to the negative (N) side of the membrane accompanied with scalar proton uptake on the N side and release on the P side. A prototypical example is nitrate respiration with formate in *Escherichia coli* performed by FdhGHI and NarGHI, both of which belong to the Mo/W-*bis*PGD family and have an opposite orientation relative to the cytoplasmic membrane[35]. In both cases, the key protein for energy conservation is a di-heme cytochrome *b* that performs transmembrane electron transfer and reduces/oxidizes the quinone pool, with the crucial difference that the quinone binding site is located on the N site for FdhI and P side for NarI. This means that menaquinone reduction by FdhGHI is accompanied with H$^+$ uptake from the N side, and menaquinol oxidation by NarGHI is accompanied with H$^+$ release to the P side.

The huge diversity in prokaryotic anaerobic respiration is expressed in similar diversity in terms of the protein complexes involved. These have been classified into different types, according to the topological organization of their quinone and

substrate-binding sites, resulting in electrogenic, electroneutral, or energy-consuming reactions[35]. Among these protein complexes, those including the QrcCD/NrfCD/PsrBC dimeric module were classified as electroneutral, either when working as quinone reductases (Type 2) or as quinol dehydrogenases (Type 6), based on the location of the quinone and substrate-binding sites on the same side of the membrane (P side), and on the fact that some of the redox reactions do not allow for energy conservation. Here we show that, in contrast to this prediction, the QrcABCD complex is electrogenic and generates a proton motive force (pmf) by transmembrane charge separation, in a redox loop mechanism with the expected $H^+/e^-$ ratio of 1. This is achieved through the uptake of protons from the N side of the membrane (cytoplasm) for reduction of MK, which binds close to the P side (periplasm), with electrons coming also from this side (Fig. 1). Using a homology-based model, a proton-conducting pathway formed by strictly conserved residues has been identified in QrcD leading from the cytoplasm to the MK binding site. This pathway is likely responsible for delivery of the protons consumed in MK reduction, and it is conserved in the closer members of the family, namely in HybB, a subunit of hydrogenase-2 (Hyd-2 or HybO-CAB) that also works as quinone reductase during respiratory metabolism[31], and in HmcC, whose physiological function and direction of electron transfer is still uncertain[1]. Thus, it seems likely that Hyd-2 conserves energy during $H_2$ oxidation coupled to quinone reduction, through uptake of protons from the cytoplasm by HybB, as observed for QrcD (and the same may happen for HmcC). This agrees with the finding that during fermentative growth on glycerol production of $H_2$ by Hyb-2 is dependent on a pmf[31]. The QrcABCD complex can also act reversibly, as shown by its essential role in syntrophic growth[18,19]. In both these situations we suggest that the pmf is essential not only to allow proton release to the cytoplasm upon quinol oxidation by HybB/QrcD, but also to provide energy for the unfavorable production of $H_2$ (through Hyd-2) or $H_2$/formate (through QrcABCD/TpI$c_3$) from quinol, by reverse electron transport.

The system composed by soluble hydrogenases/formate dehydrogenases, TpI$c_3$ and QrcABCD is an interesting biological alternative to the more standard membrane-associated hydrogenases and formate dehydrogenases that directly reduce the quinone pool through their cytochrome $b$ membrane subunit[6]. These cytochromes $b$ have quinone binding sites on the N side and use the two hemes to achieve transmembrane electron transfer against the electrochemical potential[45]. The biological solution of placing the quinone binding site on the periplasmic side and taking up protons from the cytoplasm, as observed in QrcD, obviates the need to transfer electrons against the electrochemical potential and achieves the same result in terms of charge separation. This simple solution is reminiscent of the $bd$ oxidases where the quinol and the $O_2$-reducing binuclear sites are both present on the P side, again avoiding electron transfer against the membrane potential, and protons for $O_2$ reduction are taken up from the cytoplasm[46]. Interestingly, the same structural module is found in the two subunits of the $bd$ oxidases (CydAB) as in the QrcD family: a duplicated four helix bundle. This suggests that this structural motif may be an electrogenic biological unit to perform transmembrane proton transfer from the N side to an active site on the P side.

Some members of the QrcD/NrfD/PsrC family have been proposed to operate as proton pumps[23,29,31]. We showed that the QrcABCD complex does not pump protons, even though the energetics of the reaction could in principle allow it, since there is a large drop in redox potential from the donor ($H_2$ or formate, $-420$ or $-432$ mV) to the acceptor (MK, $-70$ mV). Possible proton pumping by other members of the family will have to be tested experimentally. The QrcCD/NrfCD/PsrBC dimeric module

is found in a wide variety of prokaryotic oxidoreductase systems operating on substrates with a disparate range of redox potentials (Supplementary Table 1). This means that the free energy available from these conversions will also be quite different among the different systems, suggesting that this redox module may not always be electrogenic. In some cases the reactions may even require energy and be driven by the pmf, (as discussed above for the reverse operation of Qrc and Hyd-2), such as in the case of the PsrABC polysulfide reductase[22] or SreC sulfur reductase, which reduce low potential acceptors. In other cases, the conversions are highly favorable and clearly allow for energy conservation. From the results presented it seems likely that HybB and HmcC employ a similar redox loop mechanism for energy conservation. In other proteins of the family, although the identified residues are not conserved, it is possible that alternative residues may perform a similar function, when thermodynamics allow for energy conservation, but further studies are required to clarify this point. An intriguing respiratory complex also containing the QrcCD-like dimeric module is the reductive dehalogenase complex of *Dehalococcoides mccartyi*, composed of two hydrogenase subunits, three QrcBCD-like subunits, and two reductive dehalogenase subunits[47]. This organism contains no quinones or cytochromes so the complex apparently mediates direct electron transfer from $H_2$ to the halogenated substrate, without quinone involvement[47]. The free energy available from this conversion is quite large, and this is reported to be the only complex in *D. mccartyi* allowing for energy conservation. In the absence of quinones this process is likely to involve proton pumping by the QrcD-like subunit, driven by a redox-triggered conformational change that may also involve additional membrane subunits present in this complex such as the reductive dehalogenase membrane subunit.

Interestingly, for nine out of 13 (known) redox systems where the QrcCD-like dimeric redox module is present the substrates or pathways operate on sulfur compounds, suggesting a more intimate, and possible evolutionary, link to sulfur metabolism. In terms of sulfate respiration, the QrcABCD complex is not strictly essential, as some sulfate reducers (e.g., in Clostridia and Archaea) do not have this complex, and instead have membrane-anchored hydrogenases and formate dehydrogenases that directly reduce the quinone pool. The option of using soluble hydrogenases/formate dehydrogenases, TpI$c_3$ and QrcABCD in the Deltaproteobacteria sulfate reducers is apparently linked to a more flexible energy metabolism. In the model organism *D. alaskensis* G20 the QrcABCD complex is essential for growth on hydrogen[8] and it has also been implicated in syntrophic interactions with methanogens both in *D. vulgaris* Hildenborough and in *D. alaskensis* G20[18,19]. The Deltaproteobacteria sulfate reducers are rich in multiheme cytochromes $c$, which are responsible for intracellular[15] and extracellular electron transfer[48]. The TpI$c_3$ is the most abundant cytochrome in these organisms and serves as the periplasmic redox hub in these electron transfer pathways, while its physiological partner, the QrcABCD complex, provides the connection to the membrane quinone pool. This complex may be coupled with the QmoABC complex that is involved in electron transfer to the APS reductase in the sulfate reduction pathway[49,50] (Fig. 1).

In conclusion, this study provides an answer to the long-standing question of how energy is conserved during the dissimilatory reduction of sulfate, by showing that electron transfer from the TpI$c_3$ to the menaquinone pool, through the QrcABCD complex, is electrogenic allowing for energy conservation. Remarkably, the activity measured for this complex reconstituted in liposomes is more than 500× higher than previously reported, and approaches physiological values. In a more general context, this study provides an example of how this widespread dimeric

redox module can contribute to energy conservation in prokaryotic metabolism. It also provides an example of an electrogenic complex involved in a redox loop where the MK and substrate sites are on the same side of the membrane, defining a new type of prokaryotic respiratory system[35].

## Methods

**Liposomes preparation.** A 10 mg/ml L-α-phosphotidylcholine (PC) suspension prepared in 1 mM HEPES pH 7.3, 150 mM $K_2SO_4$ was extruded through a 0.1 μm pore size membrane. Liposomes containing menaquinone-4 (MK4) 0.5 mg/ml, were prepared dissolving MK4 and lipids in chloroform. The mixture was dried under nitrogen flux, and the process repeated twice. After chloroform evaporation, the lipid film formed was dissolved in buffer solution (10 mg PC/ml) and extruded as described above. All liposome preparations were stored at −80 °C prior to use. Incorporation of phenol red inside the liposomes was performed, as described adding 200 μM phenol red to the buffer composition before extrusion.

**Protein purification and reconstitution into liposomes.** Qrc and TpIc$_3$ were purified from membrane and soluble extract, respectively, of *D. vulgaris* Hildenborough (DSM 644)[7,51]. Membrane proteins were extracted from membranes with 2% (w/v) n-dodecyl-β-D-maltoside and the solubilised proteins were loaded on a Q-Sepharose column equilibrated with 20 mM Tris–HCl pH 7.6, 10% (v/v) glycerol, 0.1% (w/v) n-dodecyl-β-D-maltoside. A stepwise gradient of increasing NaCl concentration was applied and the fraction eluted at 250 mM NaCl was concentrated and its ionic strength lowered by ultrafiltration. This fraction underwent a second similar chromatographic step on Q-Sepharose and the heme containing fraction obtained was loaded into an IMAC-Sepharose HP column, saturated with Ni$^{2+}$, and equilibrated with 20 mM Tris–HCl pH 7.6, 0.4 M NaCl, 10% (v/v) glycerol, 0.1% (w/v) n-dodecyl-β-D-maltoside. The adsorbed proteins were eluted in a stepwise gradient with increasing imidazole concentration. Pure QrcABCD fractions were concentrated, buffer exchanged to 50 mM potassium phosphate buffer pH 7.0, 10% (v/v) glycerol, 0.1% (w/v) n-dodecyl-β-D-maltoside and frozen −80 °C prior to use. The soluble extract was purified on from Q-Sepharose HP column equilibrated in 20 mM Tris–HCl pH 7.6 and the cytochrome-containing fraction eluted before the start of the gradient was pooled and concentrated by ultrafiltration. This fraction was loaded in a SP-Sepharose HP column equilibrated in the same buffer and the adsorbed proteins were eluted in a stepwise NaCl gradient. Pure TpIc$_3$ sample was buffer exchanged to 1 mM HEPES pH 7.3, 150 mM $K_2SO_4$. All protein purification chromatographic steps were performed at 4 °C and monitored by UV–vis spectroscopy and SDS-PAGE electrophoresis.

QrcABCD was reconstituted to liposomes using Rigaud's modified method[36,39,52]. After thawing, liposome suspension was sonicated three times for 40 s and mixed with QrcABCD complex solubilized in Triton X-100, to a final concentration of 2.5 μM (0.06 mg Qrc/mg phosphotidylcholine). Valinomycin was added to the suspension to a final concentration of 100 nM, incubated for 30 min with gentle shaking at 4 °C followed by three freeze/thaw cycles. Detergent was removed by incubation with Bio-Beads SM-2 adsorbent (0.5 g/ml solution) for at least 60 min at 4 °C with gentle shaking. Supernatant suspension was removed and centrifuged to isolate proteoliposomes (213,000 × g for 60 min at 4 °C), this procedure was repeated twice when phenol red was incorporated. Finally the pellet was resuspended in the appropriate buffer 1 mM HEPES pH 7.3, 150 mM $K_2SO_4$, with or without phenol red (25 μM). Four types of liposomes were prepared in parallel: empty liposomes, Qrc proteoliposomes, MK4 liposomes, and Qrc/MK4 proteoliposomes. In order to obtain MK4 containing liposomes with comparable amounts of quinone in the lipid layer, each MK4-lipid mixture was treated equally, separated for Qrc reconstitution and MK4 liposomes preparation.

For quality control, QrcABCD proteoliposomes were dissolved in ice-cold acetone and incubated for 30 min at −20 °C. The precipitated proteins were recovered by centrifugation at 17,000 × g for 30 min at 4 °C. After acetone evaporation, samples were resuspended in buffer and analyzed by Tricine SDS-PAGE, independently stained for silver and Commassie blue followed by heme-stain (DMB method[53]) (Supplementary Fig. 2). In parallel, pure QrcABCD protein (Supplementary Data 1) and the liposome-extracted protein (Supplementary Data 2) were analyzed by mass spectrometry on a nano LC-Triple-TOFF at the Mass Spectrometry Unit (UniMS), ITQB/iBET, Oeiras, Portugal.

**Vesicles integrity.** To evaluate liposomes integrity, a membrane potential formation was detected measuring changes on Oxonol VI absorption (A$_{628nm}$ minus A$_{587nm}$) with an OLIS upgraded Aminco DW2 dual wavelength spectrophotometer at 30 °C. Valinomycin-permeabilised liposomes were incubated in 1 mM HEPES pH 7.3, 150 mM $K_2SO_4$, 1.5 μM oxonol VI, and $K^+$ gradients were initiated by the addition of KCl saturated solution (prepared in 1 mM HEPES pH 7.3, 150 mM $K_2SO_4$). Control experiments were prepared replacing KCl solution by equal volume of buffer solution.

**Proteoliposome protease hydrolysis.** Qrc proteoliposomes in 1 mM HEPES pH 7.3, 150 mM $K_2SO_4$ were mixed with protease K (0.6 μg/ml), incubated at 37 °C for 1 to 10 min and immediately transferred to ice. Protein was immediately

precipitated with ice-cold acetone and recovered by centrifugation at 17,000×g for 30 min at 4 °C. Acetone was removed, the pellet dried under nitrogen flux, resuspended in buffer and analyzed by Tricine SDS-PAGE, stained for heme (DMB method[53]) and total protein. Control experiments were performed with liposomes containing horse heart cytochrome *c* in the inside compartment, where the cytochrome was added to the lipid suspension prior extrusion as described elsewhere[39].

**Stopped-flow experiments and data analysis.** TpIc$_3$ (8–10 μM solution, $\varepsilon_{(552nm)}$ = 116 mM$^{-1}$ cm$^{-1}$) was reduced under anaerobic conditions (glove box 98% Argon, 2% $H_2$ atmosphere) with substoichiometric amounts of sodium dithionite. This procedure was followed spectroscopically to ensure no excess of reductant was present. Reduced TpIc$_3$ (approx. 80–90% reduced) was mixed with the different liposome preparations in a ratio of 1:1 (v/v) for a total reaction volume of 150 μl, in a stopped-flow instrument (KinetAsyst SF-61 DX2 from TgK Scientific) installed inside an anaerobic chamber. The temperature of the kinetic experiments was kept at 30 °C using an external circulating bath and changes in the absorption spectra were monitored by a diode array in the wavelength range 350–700 nm. TpIc$_3$ re-oxidation was followed by changes at 552 nm (specific for the reduced cytochrome α-band). Membrane potential was detected by oxonol V, recording absorbance difference, 640 nm minus 612 nm. Oxonol VI could not be used because its absorbance overlaps with that of TpIc$_3$. Phenol red protonation state was followed at 560 nm, which is an isosbestic point for TpIc$_3$. Total amount of MK4 present in each liposome preparation was determined by dissolving the prepared liposomes in absolute ethanol and quantified spectroscopically $\varepsilon_{(270nm–290nm)}$ = 14.6 mM$^{-1}$ cm$^{-1}$ [54]. For each stopped-flow experiment, the liposomes were diluted to a MK4 concentration of 20 μM, before mixing. Total Qrc concentration present in each liposome preparation was determined spectroscopically dissolving the liposomes in a 0.05% (v/v) detergent solution in the presence of excess dithionite $\varepsilon_{(552nm)}$ = 113.6 mM$^{-1}$ cm$^{-1}$ [7]. The Qrc concentration varied between 10 and 15 nM before mixing. In the end of every set of experiments, reduced TpIc$_3$ was mixed with buffer solutions containing oxygen or dithionite for further conversion of absorbance traces to TpIc$_3$ or ferrous heme concentration, using Eq. (1):

$$[\text{TpIc}_3\ \text{Fe}^{2+}] = \frac{A_{552\,nm}^{Obs} - A_{552\,nm}^{Ox}}{A_{552\,nm}^{Red} - A_{552\,nm}^{Ox}} \times [\text{TpIc}_3] \times n \qquad (1)$$

where $A^{Ox}$ and $A^{Red}$ are the absorbance at 552 nm for fully oxidized and fully reduced TpIc$_3$, respectively, $A^{obs}$ is the measured absorbance, [TpIc$_3$] the total concentration of cytochrome determined in its fully reduced state ($\varepsilon_{(552nm)}$ = 116 mM$^{-1}$ cm$^{-1}$) and $n$ the number of hemes present in the protein ($n = 4$).

All kinetic traces correspond to the average of at least seven replicates performed with at least two distinct liposome preparations. For each individual trace, the absorbance at 700 nm was used to subtract signal noise. Trace subtractions were performed only for traces acquired in the same day and with liposomes prepared in the same date, in order to minimize differences in MK4 content. Phenol red absorption at 560 nm was calibrated using tryptic hydrolysis of TAME[55]. Briefly, using the stopped-flow equipment, TAME solutions of different concentrations were mixed with a 200 μM trypsin solution prepared in 1 mM HEPES pH 7.3, 150 mM $K_2SO_4$, 200 μM phenol red (pH adjusted just before use). Ionophores as valinomycin or CCCP were incorporated in the liposomes preparation as stated or added to the prepared liposomes from concentrated solutions to a final concentration of 5 or 50 μM, respectively.

**Turnover determination from single exponential fit parameters.** To determine the $k_{cat}$ for the QrcABCD complex reconstituted in liposomes, the corrected 552 nm absorbance trace was converted to concentration of substrate [TpIc$_3$] (μM) as a function of time, and fitted with a single exponential ($A \exp^{(-kt)} + C$) (Supplementary Fig. 5).

Considering the single exponential equation for substrate evolution in time:

$$[S] = A \exp^{(-kt)} + C \qquad (2)$$

The initial velocity ($v_0$) is the symmetrical of the slope tangential to the starting point ($t = 0$), so:

$$v_0 = -\left(\frac{d[S]}{dt}\right)_{t=0} = -\left(-Ak \exp^{(-kt)}\right)_{t=0} \qquad (3)$$

$$v_0 = Ak \qquad (4)$$

Since $V_{max} = k_{cat}[E]_{total}$ and substrate saturation is guaranteed in this experiment:

$$v_0 = V_{max} = Ak \qquad (5)$$

Therefore, $k_{cat} = \frac{Ak}{[E]_{total}}$, where $k$ is the kinetic rate constant, A the exponential amplitude, both determined from the single exponential fit, and [E]$_{Total}$ = [Qrc] present in the reaction after mixing.

**Homology-based modelling of QrcCD**. The structural model of QrcCD was generated using the structure of *R. marinus* alternative complex III subunits as a template (PDB: 6F0K[29]). The model of QrcC was built using the C-terminal region (residues 796–1035) of ActB as a template. The first 8 residues of QrcC could not be modelled since they did not align well with the template. The model of the QrcD membrane subunit was based on the structure of ActC. A sequence alignment built with ClustalX 2.0 was used as input to build the model, using the automodel class implemented in Modeller[56], version 9.6, with the refinement degree set to slow. The final model corresponds to the one with the lowest value of the objective function, out of 100 generated structures.

**Phylogenetic analysis**. An all-vs-all BLAST[57] (version 2.4.0+) of the selected proteins was performed and the hits were filtered by an *E*-value cut-off of $10^{-10}$. In the second step, protein families were filtered by 25% local identity and families without relationships with other families (SreC, DmsC, SoeC, and PsrC2) were excluded from further analysis. A multiple sequence alignment was calculated in PRALINE™[58] with default parameters and a maximum-likelihood phylogeny reconstructed with IQ-tree[59] (version 1.6.2) with 1000 ultrafast bootstrap replicates, -alrt 1000 and best model selection (best model: LG+F+I+G4). Network was represented in Cytoscape[60].

## Data availability

The data supporting the findings of the study are available in this article and its Supplementary Information files, or from the corresponding author upon request.

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

## Acknowledgements

The authors would like to thank Manuela M. Pereira for helpful discussions on liposome experiments and for provision of the ActC structure prior to publication, and Miguel Teixeira for use of the dual-wavelength OLIS spectrophotometer. This work was financially supported by Fundação para a Ciência e Tecnologia (Portugal) through fellowships SFRH/BPD/84607/2012 (to A.G.D.) and SFRH/BPD/92537/2013 (to D.L.), grants PTDC/BIA-MIC/6512/2014 and PTDC/BIA-MIC/2723/2014 (to I.A.C.P.) and R&D units UID/Multi/04551/2013 (Green-IT) and LISBOA-01-0145-FEDER-007660 (MostMicro) cofunded by FCT/MCTES and FEDER funds through COMPETE2020/POCI. Funding by the WWTF (VRG15-007 to F.L.S.), BBSRC UK (Grant K00929X to T.A.C.), and the European Union's Horizon 2020 research and innovation programme (Grant agreement no. 810856) is also acknowledged.

## Author contributions

I.A.C.P. and A.G.D. conceived the study. Experimental work and data analysis was performed by A.G.D., T.C., and I.A.C.P. G.F.W. and T.A.C. were involved in proteoliposome studies. D.L. and C.M.S. were involved in protein modelling. S.N. and F.L.S. were involved in phylogenetic studies. A.G.D. and I.A.C.P. wrote the manuscript with contributions from the other authors. All authors read and approved the manuscript.

## Additional information

**Competing interests:** The authors declare no competing interests.

