## [Peer Review File · Nature Communications]

Reviewers' comments:

Reviewer #1 (Remarks to the Author):

Duarte et al. describe an analysis of electron and proton transfer catalyzed by a quinone reductase complex (QrcABCD) involved in the sulfate reduction process found in many delta proteobacteria. This study is important for a number of reasons, not least of which is that the authors propose a plausible bioenergetic mechanism to describe how this complex, whose membrane component lacks a cofactor, contributes to energy conservation. The study strongly suggests that the complex is electrogenic and that the protons for MK reduction are provided from the cytoplasmic side of the membrane, while the electrons are delivered from the periplasmic side. If correct, the system uses a Mitchell-type 'redox loop', rather than a proton-pumping mechanism to establish the proton gradient, and this would be novel. I have a few critical comments for the authors to consider:

1. The title of the manuscript is not particularly informative and does not really convey to the reader the significance of what the authors appear to have discovered. Perhaps a more 'punchy' title could be thought of that includes the novelty (being electrogenic) and importance (nevertheless redox loop-based) of the findings. The 'bacterial di-protein redox module' part will go completely over the heads of most readers and they will not stop to look more closely at the content of the manuscript, let alone read it, which would be a terrible waste because people should definitely read this paper. Moreover, this may well represent an ancient form of energy conservation, hence its widespread nature amongst anaerobes. This is completely off the top of my head, but why not something along the lines of "An electrogenically-driven redox loop in sulfate reduction reveals an ancient and widespread mechanism of energy conservation"?
2. The HybB protein of *E. coli* is part of an enzyme complex that is reversible, depending on the physiology of the cell. Can the authors make a clearer statement in the text about how this might be feasible, assuming an electrogenic mechanism similar to that proposed for Qrc?
3. A system has recently been discovered in organohalide-respiring bacteria that has a similar composition to Qrc, particularly regarding the redox protein with four FeS clusters and a cofactor-less membrane component; however, there is no evidence of quinone involvement. How would this fit with the proposed electrogenic mechanism, or could the system be 'adapted' to 'pump' protons rather than merely having a proton channel?
4. In Fig. 3C and 3D, how is it possible to accurately measure phenol red deprotonation when it forms the bulk phase? This was not completely clear.
5. While the structural model is highly plausible, it could easily be tested by making and testing a couple of variants, e.g. a D70A variant to test the proton channel or a M139F or A variant to disrupt the quinone-binding site. Analysis of these variants would provide even stronger arguments for the proposed mechanism.

Reviewer #2 (Remarks to the Author):

Duarte and colleagues present evidence that the menaquinone-reducing Qrc complex from sulfate reducers forms a redox loop (proton per electron ratio of one) during oxidation of formate or hydrogen gas via the Tplc3 multiheme cytochrome. Since QrcCD constitute a redox module of the widespread NrfD/PsrC family, these findings are highly significant and have an impact on bacterial bioenergetics in general.

The experiments are well-conceived and the results are complete, sound and convincing. The manuscript is nicely presented and the conclusions justified.

The following minor comments merely aim to improve the presentational side of the manuscript.

- 1) Line 22: The sub-clause is misleading since anaerobic reactions do not necessarily involve "low" free energy changes. This certainly is not the case for formate or hydrogen oxidation coupled to menaquinone reduction catalyzed by the Qrc complex. See also line 346.

- 2) Line 25: How about "part of the QrcABCD complex"
- 3) Lines 31/32 Misleading since different ways of operating this redox module are apparent.
- 4) Fig 1 and legend: "H₂ase" would be more appropriate than "Hase". In line 56 "2H+" could be stated.
- 5) Line 60: My view is that the "redox loop" is formed during formate (or H₂) oxidation by menaquinone and not by the coupling of the Qrc and Qmo complexes. It might well turn out in the future that menaquinol oxidation by APS forms another redox loop performed by the Qmo/Apr assembly. See also lines 415/6.
- 6) Line 70: Specify "four-cluster subunits"
- 7) Line 83: What is meant by the word "independently"?
- 8) Line 90: This rather refers to a "polysulfide reductase-like enzyme". The trouble with the *Thermus* Psr complex is that there is no biochemical evidence as to the reaction this enzyme might catalyse. As the authors stated, polysulfide reduction is not obvious, supported by primary structure difference between usual PsrCs and PsrC from *Thermus* (here called PsrC2). See also line 99.
- 9) Lines 97/98: The authors should be aware that the assumed electroneutrality is based not only on topological but also on thermodynamic considerations.
- 10) Line 409 "In these organisms": for how many organisms has this been shown?

Reviewer #3 (Remarks to the Author):

This manuscript by Duarte et al addresses the question whether and if so, how an enzyme complex involved in sulfate respiration converts an electron transfer reaction into a proton motive force. The main finding of this work is that the QrcABCD complex is electrogenic and generates a proton motive force despite the fact that it is not a proton pump and all electron transfer reactions occur on the periplasmic side of the membrane. In a redox loop mechanism the enzyme complex extracts protons from the cytoplasm with a proton to electron stoichiometry of 1. A putative channel for proton access is suggested. The proposed mechanism for QrcABCD is novel and has fundamental implications for bioenergetics in general. The manuscript is well written and of interest for readers in the bioenergetics field but also for a wider audience. Nevertheless, several issues need clarification.

The QrcABCD complex oxidizes a soluble periplasmic tetraheme cytochrome c₃ and transfers the electrons to menaquinone. The structure of the QrcABCD complex is not known but the topology can be deduced from related enzyme complexes. All redox active cofactors are situated on the periplasmic side of the membrane and the only menaquinone binding site was proposed to be also on this side of the membrane. The authors conducted several well designed experiments to show that the electron transfer reaction generates an electrochemical membrane potential by extracting protons required for menaquinone reduction from the cytoplasm. For any conclusions drawn from these experiments it is extremely important to use a highly pure enzyme preparation, especially as there are apparently no specific inhibitors (?) for this enzyme complex. A supplementary figure with a silver stained gel of a typical enzyme preparation must be included. It would be even more convincing to conduct a mass spec analysis of washed vesicles after reconstitution because contaminants might enrich in the vesicles.

Membrane potential generation was measured by following absorption changes of oxonol V. It is somewhat surprising to see the drift in the signal in the presence of valinomycin in all four samples (Suppl Fig. 3). These background rates were subtracted and each curve in Fig. 2 is the average of at least seven experiments. To get an impression of the variation in the individual experiments it might be useful to show also an overlay of the single curves (at least for Qrc/MK4) as a supplemental figure. There was also an unspecific electron transfer from the cytochrome c₃ to menaquinone which was subtracted. The difference curve showed that the enzymatic electron transfer reaction was essentially completed after 5 seconds. Nevertheless, membrane potential steadily increased over a time window of 15 seconds. How can this be explained?

Proton uptake from the cytoplasmic side was investigated by using phenol red as pH indicator. The absorption change of phenol red on the inside clearly correlated with the enzymatic electron

transfer reaction while in a different experiment the phenol red response on the outside showed no difference between vesicles with enzyme and menaquinone and vesicles only with menaquinone. The latter finding was taken to exclude a proton pumping mechanism because changes on the outside were exclusively caused by unspecific side reactions. In an experiment with phenol red in the lumen of the vesicle there was no difference in the alkalisation of the interior in the presence or absence of an uncoupler corroborating that QrcABCD is not a proton pump and that only chemical protons are involved. However, isn't it a bit surprising to see no pH equilibration with the much larger volume of the outside medium in the presence of uncoupler? Comparison with the complex I pump experiments seems not perfectly convincing because there alkalisation of the much larger outside medium by net proton consumption was monitored.

A homology model of QrcD was calculated to explore a possible access route for protons from the cytosol to the menaquinone binding site. It might be helpful to indicate the approximate boundaries of the transmembrane area in Figure 5. The distances between the two tyrosine residues in the central part to D70 and to the cluster of residues on the cytoplasmic side seem to be quite large. How does that compare to established proton conducting networks? The model was based on ActC, a subunit of the alternative complex III from *Rhodothermus marinus*. This enzyme complex was previously proposed to harbor two proton channels and to function as a redox driven proton pump. However, residues identified here to form a putative channel are not conserved in ActC. That means the putative channels in ActC and QrcD formed completely independently and the channels extend along different routes while the Q binding sites overlap? In this paragraph it should be made more clear which enzyme complexes are proposed to employ the novel redox loop mechanism described here and how wide spread it is. Maybe it would help to highlight those complexes in supplemental table 1 which lists enzyme complexes operating by different mechanisms.

Reply to Reviewer's comments

We thank all three reviewers for their very valuable comments, which were a great contribution to improve the manuscript, and also for their appreciation of this work.

Note: The revised version is submitted both with marked changes (in red, named "Related manuscript File"), and in a final unmarked version. The line numbers shown below refer first to the marked version, and after "/" to the final unmarked version.

Reviewer #1

1. *The title of the manuscript is not particularly informative and does not really convey to the reader the significance of what the authors appear to have discovered. Perhaps a more 'punchy' title could be thought of that includes the novelty (being electrogenic) and importance (nevertheless redox loop-based) of the findings. The 'bacterial di-protein redox module' part will go completely over the heads of most readers and they will not stop to look more closely at the content of the manuscript, let alone read it, which would be a terrible waste because people should definitely read this paper. Moreover, this may well represent an ancient form of energy conservation, hence its widespread nature amongst anaerobes. This is completely off the top of my head, but why not something along the lines of "An electrogenically-driven redox loop in sulfate reduction reveals an ancient and widespread mechanism of energy conservation"?*

Author's reply: We agree with the Reviewer that the title was not very informative, and thank very much for this suggestion, which we adopted with a small modification. The new title is indeed much better and more likely to catch the reader's attention. We also agree this may be a particularly ancient mechanism of energy conservation and changed the abstract and text (lines 32/30, 93-94/90-91 and 339/337) to further stress this.

2. *The HybB protein of E. coli is part of an enzyme complex that is reversible, depending on the physiology of the cell. Can the authors make a clearer statement in the text about how this might be feasible, assuming an electrogenic mechanism similar to that proposed for Qrc?*

Author's reply: We have included such a statement in lines 407-410/404-408. In addition, we added a comment on the reversibility of the Qrc and Hyd-2 complexes, which was missing but is also very pertinent in this discussion (lines 410-414/408-412).

3. *A system has recently been discovered in organohalide-respiring bacteria that has a similar composition to Qrc, particularly regarding the redox protein with four FeS clusters and a cofactor-less membrane component; however, there is no evidence of quinone involvement. How would this fit with the proposed electrogenic mechanism, or could the system be 'adapted' to 'pump' protons rather than merely having a proton channel?*

Author's reply: This complex is indeed very interesting, since in this case proton uptake for quinone reduction is not possible. We agree with the Reviewer's suggestion that the system has likely been "adapted" for proton pumping. We think this maybe be related

with the presence of additional membrane subunits, namely the reductive dehalogenase membrane subunit, which may be involved in a redox-triggered conformational change leading to proton pumping, together with the QrcD-like subunit. This discussion has been included in lines 445-453/443-451, and we thank the Reviewer for raising this interesting point.

4. *In Fig. 3C and 3D, how is it possible to accurately measure phenol red deprotonation when it forms the bulk phase? This was not completely clear.*

Author's reply: This is possible because we used a considerable amount of liposomes, resulting in a significant concentration of protons taken up from the outside solution for the unspecific reduction of MK4 by the cytochrome. To quantify this we calculated the concentration of protons taken up outside the liposomes, as well as the concentration of heme oxidized in the process, from Figures 3c and 3d, using the same method as used for Fig. 4. The results, shown in new Supplem. Fig. 9, reveal that about 4 μ M of protons are consumed outside, which is within the range that can be detected by phenol red, as shown by the calibration curve under the same conditions (now Suppl. Fig. 7). This means that the concentration of protons taken up was high enough to be detected by phenol red, enabling this experiment. As expected, the proton/electron ratio of the unspecific process, as determined from the fit of the experimental data, is also approximately 1. Since two Reviewers asked the same question, we added this discussion to the results section to clarify this point (lines 261-266/259-264).

5. *While the structural model is highly plausible, it could easily be tested by making and testing a couple of variants, e.g. a D70A variant to test the proton channel or a M139F or A variant to disrupt the quinone-binding site. Analysis of these variants would provide even stronger arguments for the proposed mechanism.*

Author's reply: We obviously agree with the Reviewer that these are very important experiments to perform, and indeed they are planned to start in the near future. However, they will unfortunately take a large amount of time, as they will first require setting up an expression system for the Qrc complex (still not available) to allow modifications in the QrcD subunit. From our experience with other *D. vulgaris* proteins, this is likely to be a lengthy process that will require either expression in *E. coli* or more likely in a *D. vulgaris* deletion mutant for the Qrc complex, which has still not been produced. On top of having to express variants of the Qrc complex, we will in addition need considerable time for the liposome experiments as described here, which are also quite difficult. For these reasons we opted to publish the current results as they are, since we believe that they represent a really important step forward in our understanding of bacterial anaerobic metabolism. These results are important not only to help understand sulfate reduction, but also a range of other respiratory metabolisms, and for this reason we believe they should be shared with the community sooner, rather than later.

Reviewer #2 (Remarks to the Author):

Duarte and colleagues present evidence that the menaquinone-reducing Qrc complex from sulfate reducers forms a redox loop (proton per electron ratio of one) during oxidation of formate or hydrogen gas via the Tplc3 multihaem cytochrome. Since QrcCD constitute a redox module of the widespread NrfD/PsrC family, these finding are highly significant and have an impact on bacterial bioenergetics in general.

The experiments are well-conceived and the results are complete, sound and convincing. The manuscript is nicely presented and the conclusions justified.

The following minor comments merely aim to improve the presentational side of the manuscript.

1) Line 22: The sub-clause is misleading since anaerobic reactions do not necessarily involve “low” free energy changes. This certainly is not the case for formate or hydrogen oxidation coupled to menaquinone reduction catalyzed by the Qrc complex. See also line 346.

Author’s reply: The Reviewer is of course correct. We meant “low” relative to aerobic respiration. Since the abstract is already at the word limit we inserted only the word “often” (“In anaerobic energy metabolism, where redox conversions **often** involve low free energy changes,...”), which we believe make the sentence more accurate. In line 346 (now line 375/373) we believe the sentence is correct since it explicitly states that in anaerobic respiration the free energy changes are smaller than in aerobic respiration.

2) Line 25: How about “part of the QrcABCD complex”

Author’s reply: We believe that adding “part of the QrcABCD complex” would make the sentence incorrect since the sentence is about respiratory complexes and not about the dimeric redox module per se. We hope the Reviewer agrees.

3) Lines 31/32 Misleading since different ways of operating this redox module are apparent.

Author’s reply: We do not really understand why the Reviewer finds the sentence misleading. We are saying that these results provide evidence of how the module “**may**” operate in other contexts, and so we are not excluding other modes of operation. We presume the Reviewer is referring to the possibilities that in other respiratory systems the redox module may not be electrogenic, may be energy-driven or may be involved in proton pumping. We believe this manuscript also provides important new insight into these possibilities in the different systems

(which will have to be studied individually), specially after the changes made during revision, so we think the sentence is correct. We also added a new comment to the text (lines 436-437/434-435), which along with other sentences introduced in the revised version, make the several possibilities clearer to the reader. This is impossible to do in the abstract within the word limit.

4) *Fig 1 and legend: "H2ase" would be more appropriate than "Hase". In line 56 "2H+" could be stated.*

Author's reply: Altered as suggested.

5) *Line 60: My view is that the "redox loop" is formed during formate (or H2) oxidation by menaquinone and not by the coupling of the Qrc and Qmo complexes. It might well turn out in the future that menaquinol oxidation by APS forms another redox loop performed by the Qmo/Apr assembly. See also lines 415/6.*

Author's reply: We deleted "forming a redox loop". We also modified the sentence in previous lines 415-416 to "This complex may be coupled with the QmoABC complex" (now line 468/466).

6) *Line 70: Specify "four-cluster subunits"*

Author's reply: Sentence modified to "electron transfer subunits containing four Fe-S clusters" (now line 72/70)

7) *Line 83: What is meant by the word "independently"?*

Author's reply: We meant that the redox module is operating alone (as in the NrfCD or MccCD systems) and not as part of a complex with other subunits. We inverted the order of the sentence and replaced "independently" by "alone" in the hope that the sentence is now clearer (now line 86-87/83-84).

8) *Line 90: This rather refers to a "polysulfide reductase-like enzyme". The trouble with the Thermus Psr complex is that there is no biochemical evidence as to the reaction this enzyme might catalyse. As the authors stated, polysulfide reduction is not obvious, supported by primary structure difference between usual PsrCs and PsrC from Thermus (here called PsrC2). See also line 99.*

Author's reply: Indeed, there is no biochemical evidence that the enzyme from *Thermus thermophilus* is a real polysulfide reductase, and the distance to the *Wolinella* PsrC suggests that the substrate may be different. We modified the sentence to make this clearer, and in previous line 99 (now line 96/93) and line 107/103 used PsrC2.

9) *Lines 97/98: The authors should be aware that the assumed electroneutrality is based not only on topological but also on thermodynamic considerations.*

Author's reply: We agree and to make this point clearer we modified the sentence that now reads: "These proteins have been assumed to be electroneutral due to the location of the quinone-binding site, and because in some cases thermodynamics precludes energy conservation."

10) Line 409 "In these organisms": for how many organisms has this been shown?

Author's reply: This has only been shown for a single organism, whereas the role in syntrophy has been shown for two organisms. The sentence was modified to state this specifically (now line 462-464/460-462).

Reviewer #3:

This manuscript by Duarte et al addresses the question whether and if so, how an enzyme complex involved in sulfate respiration converts an electron transfer reaction into a proton motive force. The main finding of this work is that the QrcABCD complex is electrogenic and generates a proton motive force despite the fact that it is not a proton pump and all electron transfer reactions occur on the periplasmic side of the membrane. In a redox loop mechanism the enzyme complex extracts protons from the cytoplasm with a proton to electron stoichiometry of 1. A putative channel for proton access is suggested. The proposed mechanism for QrcABCD is novel and has fundamental implications for bioenergetics in general. The manuscript is well written and of interest for readers in the bioenergetics field but also for a wider audience. Nevertheless, several issues need clarification.

The QrcABCD complex oxidizes a soluble periplasmic tetraheme cytochrome c3 and transfers the electrons to menaquinone. The structure of the QrcABCD complex is not known but the topology can be deduced from related enzyme complexes. All redox active cofactors are situated on the periplasmic side of the membrane and the only menaquinone binding site was proposed to be also on this side of the membrane. The authors conducted several well designed experiments to show that the electron transfer reaction generates an electrochemical membrane potential by extracting protons required for menaquinone reduction from the cytoplasm.

1. For any conclusions drawn from these experiments it is extremely important to use a highly pure enzyme preparation, especially as there are apparently no specific inhibitors (?) for this enzyme complex. A supplementary figure with a silver stained gel of a typical enzyme preparation must be included. It would be even more convincing to conduct a mass spec analysis of washed vesicles after reconstitution because contaminants might enrich in the vesicles.

Author's reply: Indeed there are no specific inhibitors. We did not include a gel of the isolated complex in the initial version of the manuscript because this has been reported before (ref. 7). We have now included Coomassie, Silver-stained and heme-stained gels for the complex as purified, and after its extraction from liposomes (new

Supplem Fig. 2). Heme staining is required for better visualization of the QrcA subunit, which does not stain with Coomassie, and poorly with Silver. These gels show that the complex has only minor contaminants, even after extraction from the liposomes.

In addition, as suggested, we performed mass spec analysis of protein as-isolated and extracted from washed vesicles (see attached file). The results show that the Qrc subunits are by far the major proteins present. Note that a smaller number of peptides is detected for QrcA, because being a multiheme cytochrome *c*, many peptides are cross-linked with heme, and so are not identified. Due to the high sensitivity of mass spec, several other proteins are also detected. The ATP synthase, highly abundant in cells, is a small contaminant, but this will not be active or affect the experiments in the absence of ADP+Pi or ATP.

2. Membrane potential generation was measured by following absorption changes of oxonol V. It is somewhat surprising to see the drift in the signal in the presence of valinomycin in all four samples (Suppl Fig. 3). These background rates were subtracted and each curve in Fig. 2 is the average of at least seven experiments. To get an impression of the variation in the individual experiments it might be useful to show also an overlay of the single curves (at least for Qrc/MK4) as a supplemental figure.

Author's reply: We have now included the raw data for the individual experiments for all four types of liposomes (new Suppl. Fig. 6, referred in the text in line 174/171).

*3. There was also an unspecific electron transfer from the cytochrome *c*3 to menaquinone which was subtracted. The difference curve showed that the enzymatic electron transfer reaction was essentially completed after 5 seconds. Nevertheless, membrane potential steadily increased over a time window of 15 seconds. How can this be explained?*

Author's reply: This is explained by the fact that we used oxonol V for these experiments, which is characterized by having a slow response time due to low mobility within the lipidic phase (see Clarke & Appell 1989 Biophys Chem 34, 225). We could not use oxonol VI since its absorbance overlaps with that of the Tplc₃. Due to the slow response of oxonol V, its absorbance changes are delayed relative to electron transfer, and so these experiments cannot be viewed in a quantitative perspective, but rather in a qualitative one. For quantification we used the experiments with phenol red. We have included a comment in the manuscript to make this point clearer (lines 177/174 and 556-557/551-552).

4. Proton uptake from the cytoplasmic side was investigated by using phenol red as pH indicator. The absorption change of phenol red on the inside clearly correlated with the enzymatic electron transfer reaction while in a different experiment the phenol red response on the outside showed no difference between vesicles with enzyme and menaquinone and vesicles only with menaquinone. The latter finding was taken to exclude a proton pumping mechanism because changes on the outside

were exclusively caused by unspecific side reactions. In an experiment with phenol red in the lumen of the vesicle there was no difference in the alkalinisation of the interior in the presence or absence of an uncoupler corroborating that QrcABCD is not a proton pump and that only chemical protons are involved. However, isn't it a bit surprising to see no pH equilibration with the much larger volume of the outside medium in the presence of uncoupler? Comparison with the complex I pump experiments seems not perfectly convincing because there alkalinisation of the much larger outside medium by net proton consumption was monitored.

Author's reply: We admit that we were also puzzled with this result for quite a while. However, several reports in the literature show that phenol red detects the consumption/production of scalar protons even in the presence of CCCP, which only inhibits/equilibrates the pumped protons. This suggests to us that there may be a significant difference between the time scales of proton uptake for chemical conversion (scalar protons) and the pumping of protons, such that CCCP only inhibits/equilibrates the latter. It must be noted that in these experiments the response rate of phenol red is measured, and not the direct rate of proton uptake. Examples of studies where this is observed include, not only the two studies mentioned in the text (regarding the Complex I study we were referring to the experiment in Fig. 5, which is identical to ours, and where the increase in phenol red absorption corresponding to the scalar protons is still observed in the presence of FCCP), but also a study on nitric oxide reductase where a similar increase in phenol red absorption was observed in the presence or absence of FCCP, in an experiment very similar to ours, and in the same timescale (Fig 1 of Reimann et al 2007 Biochem Biophys Acta 1767, 362), or a study on cytochrome c oxidase in which scalar proton uptake was detected in the presence of valinomycin/CCCP, as in our study (Fig. 6 in Vilhjálmsson et al 2015 Scient Rep 5, 12047).

5. A homology model of QrcD was calculated to explore a possible access route for protons from the cytosol to the menaquinone binding site. It might be helpful to indicate the approximate boundaries of the transmembrane area in Figure 5. The distances between the two tyrosine residues in the central part to D70 and to the cluster of residues on the cytoplasmic side seem to be quite large. How does that compare to established proton conducting networks?

Author's reply: We modified Fig. 5 to include the approximate membrane boundaries.

The distances between Y110 and D70, and Y150 and E157 in the model are 8.8 and 9.9 Å, respectively. These distances are indeed large, but not outside distances found in other proton channels, and are most likely bridged by water molecules that mediate the transfer of protons between these residues. As an example, high resolution X-ray structures of cytochrome c oxidase revealed the existence of a large number of water molecules inside the D channel (Iwata et al 1995 Nature, 376, 660; Qin et al 2009 Biochemistry 48, 5121). Water molecules are also present in the K channel of this protein, although in a smaller amount. These structures also showed that there are gaps between some of the residues that comprise these channels,

which in some cases are in order of 8-10 Å. Also, the potential proton transfer pathways in the *bd* oxidase from *Geobacillus thermodenitrificans*, proposed on the basis of its X-ray structure, contain gaps of this magnitude between protonable residues (Safarian et al 2016 Science 352, 583). Therefore, it is common to find proton pathways with large distances between some of the residues, which are usually bridged by water molecules that mediate proton transfer. A comment was inserted in the text (lines 295-296). Two further conserved residues are also likely part of the quinone-binding site (E142) and proton channel (H199).

6. The model was based on ActC, a subunit of the alternative complex III from Rhodothermus marinus. This enzyme complex was previously proposed to harbor two proton channels and to function as a redox driven proton pump. However, residues identified here to form a putative channel are not conserved in ActC. That means the putative channels in ActC and QrcD formed completely independently and the channels extend along different routes while the Q binding sites overlap?

Author's reply:

Indeed the residues proposed to form the proton channel were found to be conserved only in the proteins more closely related to QrcD, namely HybB and HmcC. Nevertheless, the proposed ActC cytoplasmic half-channel is in the same region as that found in QrcD (new Supplem. Fig. 11), although most of the protonable residues are not conserved nor in the same positions, with the exception of Y150 (QrcD numbering), which is conserved between the two proteins. We also note that in QrcD the proton pathway goes up to the quinone-binding site and is connected to this site by D70, whereas in the case of the ActC cytoplasmic half-channel there is apparently no connection to the quinone-binding site. ActC is proposed to have another half-channel connecting to the periplasmic side, which we could not observe in QrcD. So, we do not believe that the two channels formed independently, but rather that they evolved separately from a common ancestor to have different mechanisms of energy conservation. We modified the text to comment on this point (lines 297-305/295-303).

7. In this paragraph it should be made more clear which enzyme complexes are proposed to employ the novel redox loop mechanism described here and how wide spread it is. Maybe it would help to highlight those complexes in supplemental table 1, which lists enzyme complexes operating by different mechanisms.

Author's reply: We are not sure to which paragraph the Reviewer is referring to. In lines 360-362/358-360 we already indicate that the residues proposed to form the quinone binding site and the H⁺ transfer pathway in QrcD are conserved in HybB and HmcC, the two proteins that are phylogenetically more closely related, suggesting that they use the same redox loop mechanism. In the Discussion, additional comments were inserted to clarify this point (lines 440-444/438-442). Regarding the other proteins, we cannot make definite comments, as lack of conservation of the quinone-binding and/or proton channel residues does not necessarily imply a different mechanism. It is possible that different residues are performing the same

function, and only a detailed study of each protein will reveal whether a redox loop may be present or not.

REVIEWERS' COMMENTS:

Reviewer #1 (Remarks to the Author):

The authors have done an excellent job in dealing with the comments/ questions raised by the reviewers. I have no further comments apart from one extremely minor editorial issue: On line 359 of the revised manuscript the word should be 'preferentially'. 'Preferably' is usually used in reference to people and not objects.

Reviewer #3 (Remarks to the Author):

The manuscript is now ready for publication.